# Complex fault interaction controls continental rifting

John B. Naliboff [1,2], Susanne J.H. Buiter [1,3], Gwenn Péron-Pinvidic[1], Per Terje Osmundsen[1,4,5] & Joya Tetreault[1,6]

Rifted margins mark a transition from continents to oceans and contain in their architecture a record of their rift history. Recent investigations of rift architecture have suggested that multiphase deformation of the crust and mantle lithosphere leads to the formation of distinct margin domains. The processes that control transitions between these domains, however, are not fully understood. Here we use high-resolution numerical simulations to show how structural inheritance and variations in extension velocity control the architecture of rifted margins and their temporal evolution. Distinct domains form as extension velocities increase over time and deformation focuses along lithosphere-scale detachment faults, which migrate oceanwards through re-activation and complex linkages of prior fault networks. Our models demonstrate, in unprecedented detail, how faults formed in the earliest phases of continental extension control the subsequent structural evolution and complex architecture of rifted margins through fault interaction processes, hereby creating the widely observed distinct margin domains.

[1] Team for Solid Earth Geology, Geological Survey of Norway, Trondheim 7040, Norway. [2] Department of Earth and Planetary Sciences, University of California, Davis, CA 95616, USA. [3] The Centre for Earth Evolution and Dynamics, University of Oslo, Oslo 0315, Norway. [4] The University Centre in Svalbard, Longyearbyen 9171, Norway. [5] Department of Geosciences, University of Oslo, Oslo 0371, Norway. [6] Exploro AS, Trondheim 7014, Norway. Correspondence and requests for materials should be addressed to J.B.N. (email: jbnaliboff@ucdavis.edu)

Early models of continental rifting and breakup invoked one or two phases of relatively simple symmetric[1, 2] or asymmetric[3, 4] extensional deformation to explain first-order patterns of normal faulting, crustal thinning and sedimentary records. Offshore geophysical data and drill cores acquired over the past 30 years have, however, systematically revealed a structural complexity that challenges such simple models of continental rifting. Tectonic interpretations[5–11] based on these recently acquired data sets suggest that rifting occurs through multiple distinct phases of deformation that migrate oceanwards and are characterized by different structural, sedimentary and magmatic processes. Although rifted margins show significant along-strike geometrical variations and differ from each other in many details, observations show that they also contain numerous first-order similarities that point to a common set of evolutionary phases[12] (Fig. 1).

The earliest phase of preserved deformation ('stretching phase', Fig. 1a) is thought to represent the onset of continental extension. Distributed faulting, primarily recorded in the upper crust, and relatively uniform lithospheric thinning accommodate small amounts of total extension[12]. A transition to a phase of significant crustal thinning and lithospheric necking ('thinning phase', Fig. 1b) occurs when ductile layers thin sufficiently to allow brittle coupling between the lower crust and upper mantle. This accompanies rapid strain localization onto large-offset detachment faults located in the mid-to-upper crust or lower crust-to-upper mantle[7, 12]. Upon embrittlement of the entire crust, deformation migrates oceanward and further localizes onto concave downward detachment faults cutting through the entire lithosphere ('hyperextension and exhumation phase', Fig. 1c). Hyperextension of the crust, mantle exhumation and magmatism may occur during this phase, which directly precedes continental breakup and mid-ocean ridge spreading (Fig. 1d).

This conceptual model of progressive strain localization through time successfully explains first-order rifted margin features[12], but leaves key differences in margin width and asymmetry, rift abandonment and distal domain faulting patterns to be explained. Numerical simulations of continental extension indicate that many of these differences could reflect the lithosphere's initial and evolving rheological structure[7, 13–17], extension velocity[16–19], structural inheritance[15, 20], and periods of tectonic quiescence[21]. While introducing more complexity, these models do not explain the common character of the deformational domains observed among different rifted margins worldwide nor the controls on the temporal transitions between the specific deformation phases.

Using high-resolution numerical models of continental extension, we produce for the first time the full sequence of rifted margin deformation phases and their associated fundamental structural features. Our models of continental extension were developed using the thermo-mechanical tectonics code SULEC v. 6.0, which approximates ductile flow and brittle failure (i.e., shear zone formation here referred to as faulting), respectively, with viscous creep and plasticity (see Methods and Supplementary Information). Three elements are key to our models: First, the numerical resolution (250 m) is 2–8× higher than most margin studies. Second, deformation localizes on distributed heterogeneities with rapid strain weakening instead of being forced with a single zone of weakness as commonly prescribed. This allows us to achieve the distributed faulting of the initial 'stretching' phase of continental extension. Third, we include the commonly observed increase in extension rates towards continental breakup[16, 19, 22]. This allows the models to transition to the thinning phase and produces time-dependent deformation patterns that capture widely observed rifted margin structural relationships.

The models also reveal deformation in the thinning and hyperextension-exhumation phases is accommodated by both new faults and inherited stretching phase faults, which evolve through a combination of fault reactivation, abandonment, development, and incision. This process of complex fault interaction and the resulting patterns of finite-deformation are consistent with detailed observations of rifted margin structure and provide new constraints for the interpretation of rifted margin evolution.

## Results

**Modeled phases of continental rifting.** During the initial stage of extension, we find that applied extension velocities $< \sim 5 \, \text{mm yr}^{-1}$ permit uniform lithospheric thinning accompanied by distributed faulting in the brittle domains of the crust, which are decoupled by weak ductile crustal layers (Supplementary Fig. 4). Significant surface topography develops (1–2 km) along grabens and half-grabens spaced laterally by ca km's or 10's of km (Fig. 2a). These features, combined with minimal Moho displacement, directly correspond to the first-order observations of the conceptual stretching phase (Fig. 1a). In the North Atlantic region, various observations suggest the initial stretching phase of extension may have lasted for 10's Myr or longer[19]. The stretching phase in our numerical experiments may persist over these time scales for relatively low extension rates ($<2 \, \text{mm yr}^{-1}$). Notably, the associated strain rates ($< 10^{-8} \, \text{s}^{-1}$) are consistent with the range of observed strain rates in areas of distributed continental extension[23]. At $1 \, \text{mm yr}^{-1}$, distributed faulting in the decoupled upper crust, lower crust, and mantle continues for 50 Myr (Fig. 2a), at which point the ductile layers thin sufficiently to initiate coupling of deformation through the entire crust and uppermost mantle.

Estimates of extension rates during continental rifting[18, 19, 22] suggest velocity increases with time, consistent with strain localization and weakening towards the future breakup location[19]. In agreement with this observation, we find that a transition from distributed to localized deformation in the thinning phase requires a velocity increase to at least $5 \, \text{mm yr}^{-1}$ in our models (Fig. 2b). This reflects that advective heating of the lithosphere needs to exceed conductive cooling in order to allow strain localization in the lithosphere, rather than a continuing pattern of upper-plate rift migration[18]. Consistent with observations in the Atlantic[12] (Fig. 1b), deformation in the thinning phase in our models localizes in a narrow region (~50 km width) along large-offset detachment faults and smaller connecting faults, which by 64 Myr extend through the lithosphere. Further extension for 10 Myr leads to enhanced strain localization, rapid thinning of the crust and subsequent exhumation of the mantle lithosphere (Fig. 2c), matching the observed deformation sequence in the hyperextension and exhumation phase (Fig. 1c). Our final margin architecture (Fig. 2d) contains the key structural features characteristic of the proximal, necking, distal, and outer domains (Fig. 1d).

**Evolution of finite strain and active deformation.** Our simulations reveal that faulting during the earliest phases of extension exerts a key control on the complex evolution of deformation during the latter stages of continental rifting. Finite strain patterns (accumulated plastic deformation, Figs. 2a–d, 3) reveal complex patterns of fault development, deactivation, and linkage during the thinning and hyperextension-exhumation phases. This evolution of the fault network within the rift zone, as opposed to deformation along a single long-lived detachment fault, occurs through a combination of new fault development and reactivation of distributed normal faults developed in the initial stretching phase (Fig. 2a). New crustal-scale faults link stretching phase

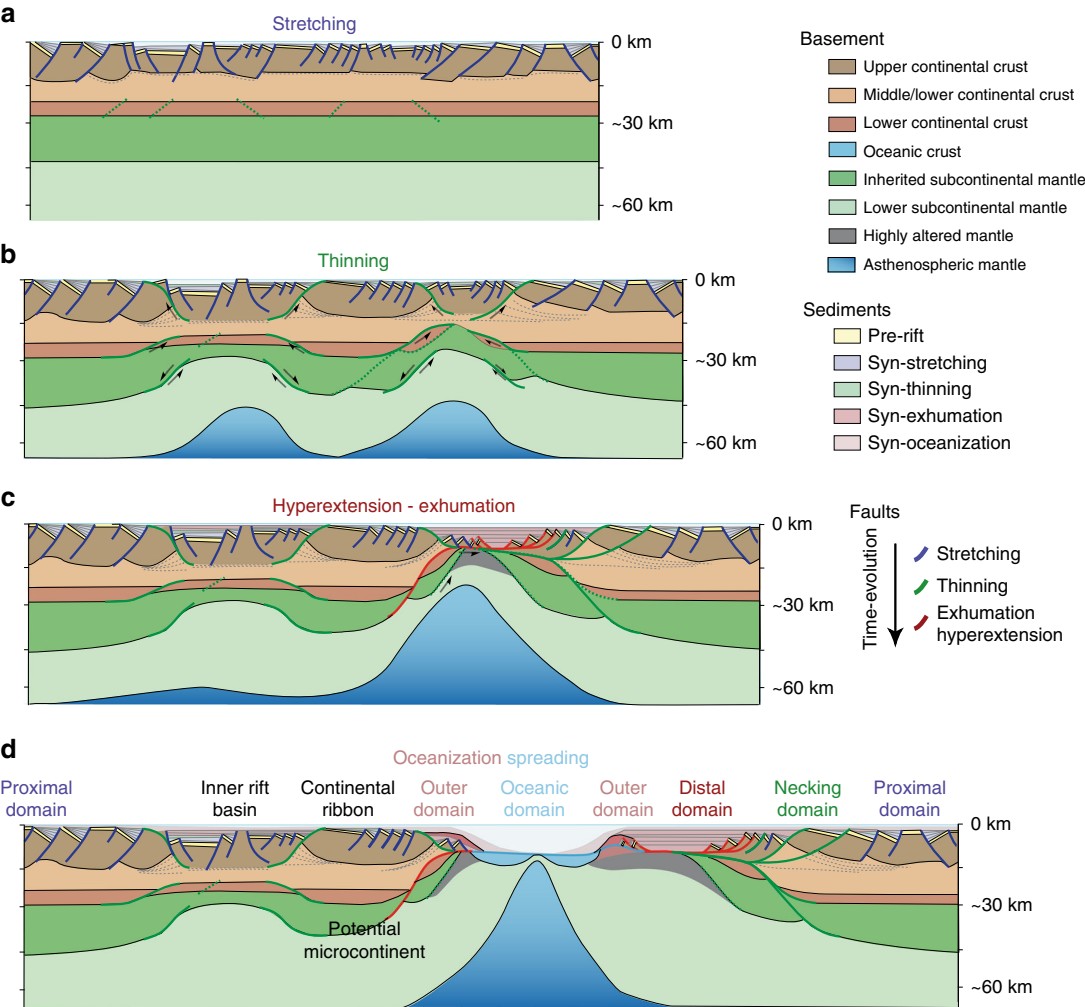

**Fig. 1** Schematic model of the phases of rifted margin formation. **a** In the stretching phase, extension is accommodated by distributed faulting in the upper crust, resulting in a distributed reduction of crustal thickness. **b** In the thinning phase, deformation localizes in focused areas characterized by deformation coupling, formation of lithospheric-scale shear zones and uplift of the asthenosphere. **c** In the hyper-extension and exhumation phases, the crust is thinned to such a degree that it behaves in a brittle-only manner. Large-scale detachment faults that cut through the lithosphere may bring water down to sub-crustal depths, thus initiating serpentinization. **d** The last phase of oceanization is characterized by crustal and lithospheric breakup and formation of oceanic crust. The margin domains link to their phase of formation as described in the text. The schematic model is reprinted with minor modifications from Péron-Pinvidic, et al.[12]

faults segments in the upper crust and lower crust (or mantle) optimally oriented to accommodate faulting (Figs. 3–4, Supplementary Movie 1).

Such comparisons between accumulated and active zones of deformation elucidate this process of fault deactivation, new fault development, reactivation, fault incision, and the resulting highly-complex cross-cutting relationships.

After 10 Myr of rapid extension (5 mm yr$^{-1}$) in the thinning phase (60 Myr total extension), initially decoupled zones of normal faults in the upper crust, lower crust, and mantle link near the model center (Fig. 3a). The corresponding active deformation field (Fig. 3e) reveals a dominant set of conjugate shear zones aligned along these pre-existing structures, with additional stretching-phase faults accommodating smaller amounts of deformation nearby.

Further extension (66 Myr, Fig. 3b) produces a distinct ramp-flat-ramp fault geometry as specific fault segments rotate from high angles to near-horizontal in the upper crust and along the Moho at the model center (region of maximum crustal thinning). The active deformation field (Fig. 3f), however, reveals that two moderately dipping detachment faults accommodate the majority

of extension and bound the horizontal (ramp) section of the primary fault structure. The active fault segments link through both structures that were recently active (Fig. 3a–e), as indicated by the zones of highest accumulated plastic strain (Fig. 3b), and newly reactivated stretching phase faults. As extension proceeds to hyperextension and mantle exhumation, significant rotation, stretching and flattening of pre-existing faults occurs (Fig. 3c, d) within the highly thinned crust of the necking, distal and outer domains (Fig. 1d).

**Interpretation of structural relationships**. This evolution of the fault network through the thinning, hyperextension, and exhumation phases (Figs. 3 and 4, Supplementary Movie 1) produces complex cross-cutting relationships, as new fault development and reactivation progressively incise and offset both deactivated and active faults. Superimposing active deformation on the accumulated deformation field (Fig. 4, Supplementary Movie 1) and tracking the evolution of distinct fault segments (F1–F4 in Fig. 4) clearly illustrates these relationships. At 63 Myr fault segments F1 and F4 are active, while segments F2 and F3 are

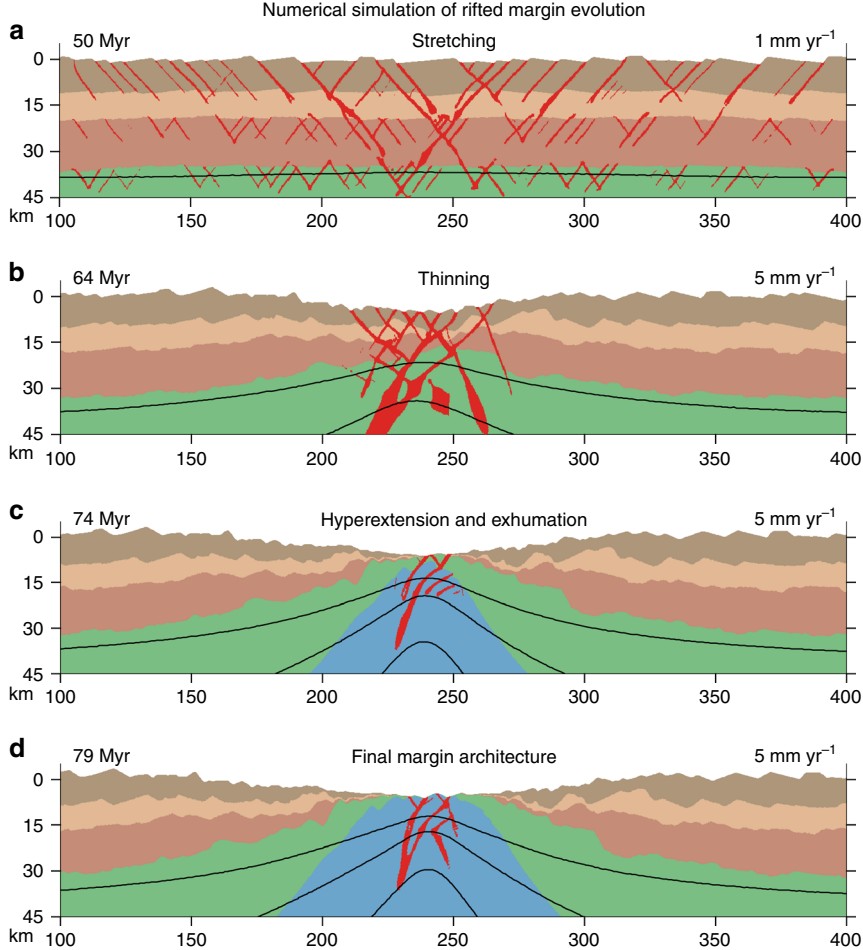

**Fig. 2** Modeled phases of rifted margin formation. Numerical model showing phases of continental extension and rifted margin formation that correspond to the conceptual phases of Fig. 1a–d. The initial model contains random small-scale disturbances in its brittle strength to help localize deformation. Images show a 300 by 45 km subset of the total model domain of 500 by 110 km. *Black lines* mark temperature contours of 600, 900, and 1200 °C. **a** 50 Myr of slow continental extension at 1 mm yr$^{-1}$ causes distributed shear zone formation throughout the crust and upper mantle lithosphere. **b–d** An increase in velocity to 5 mm yr$^{-1}$ progressively localizes deformation towards the model center in the thinning, hyperextension-exhumation and final breakup stage. *Red regions* highlight strain-rates above $10^{-15.5}$ s$^{-1}$ **a**, $10^{-14.5}$ **b**, and $10^{-14}$ s$^{-1}$ **c, d**

inactive (e.g., abandoned) and cross-cut by segment F4. Further extension to 64 Myr reveals the progressive deactivation of segment F1, while segments F2 and F3 reactivate and link to form a single active strand. This strand incises segment F4, which remains active in the footwall of segment F2–F3 but is progressively deactivated in the hanging wall of F2–F3. Notably, the linking and reactivation of segments F2–F3 initiated with deformation along F2 that connected to a fault within the mantle lithosphere. In other cases upper crustal faults may reactivate independently of faults within the lower crust and mantle lithosphere, but at this stage of deformation (thinning, Fig. 2b) the fault network evolution appears largely controlled by changes in the Moho structure (Supplementary Movie 1).

**Comparison between modeled and observed rifted margin structures.** The finite and active deformation patterns also reveal many structures commonly observed within rifted margins. Ramp-flat and core complex (concave downward) fault structures, preserved primarily in the necking zone, correspond directly to similar structures observed in the necking and outer domains of the mid-Norwegian margin[9, 24] (Fig. 5a–c). As in interpretations of high-resolution seismic data from the mid-Norwegian margin[9] (Fig. 5c, d), the ramp-flat and core complex structures (Fig. 5a) form as new, steeper detachments incise or

reactivate portions of pre-existing structures, which are subsequently rotated horizontally and vertically warped to form antiformal dome structures by basement (mid-lower crust, mantle) uplifted along the footwall (Figs. 3–5, Supplementary Movie 1). Notably, stretching of the necking and distal domains also contributes to abandoned faults rotating to sub-horizontal angles. Also consistent with detailed observations from seismic line GMNR94-103 (Fig. 5c), active deformation in the distal domain (exhumation stage) occurs along steeper structures penetrating well into the mantle lithosphere, which successively incise the rotated and abandoned sub-horizontal faults (Figs. 3g, h and 4, 5b–d). However, structures extending into the mantle lithosphere are not clearly visible in data from line OLW02-232 (Fig. 5d). The numerical results also show both seaward- and continent-dipping structures, while seismic line GMNR94-103 only reveals seaward-dipping structures. The lack of these structures in the seismic lines may simply reflect their steep-dip and the quality of data in the uppermost mantle between 10 and 14 s-twt. Also, we note that landward dipping faults are observed along other rifted margins, such as Namibia and Brasil.

## Discussion

The process of fault rotation and abandonment outlined in our experiments at first-order matches one proposed model for

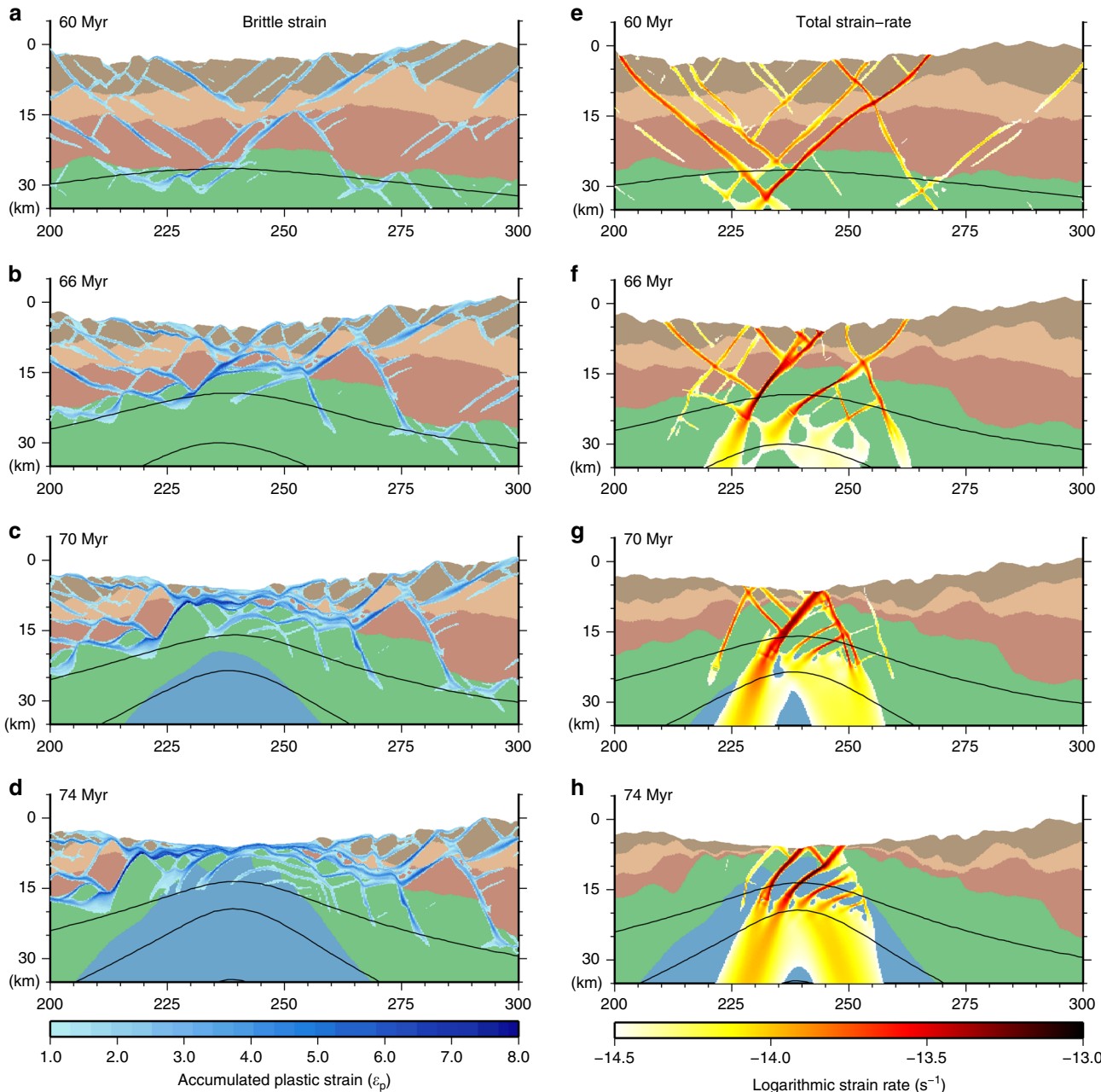

**Fig. 3** Evolution of finite strain and active deformation. Detailed evolution of the transition from stretching to hyperextension and exhumation in the numerical experiments. Each panel contains the compositional fields with accumulated plastic (brittle) strain **a**–**d** or logarithmic strain-rate **e**–**h** superimposed. As extension proceeds, high-angle faults formed in the early stretching phase are rotated sub-horizontally and stretched, while active shear zones remain at high angles

low-angle normal fault development[25]. However, interpretation of these results with respect to the debate on low-angle normal fault seismicity[26] should be considered within the context of our numerical assumptions. For example, weakening the friction angle to lower values[14, 16], including elastic effects[27] or the presence of fluids[26] may also promote accommodation of deformation on lower angle structures. Similarly, higher numerical resolutions and constitutive relationships associated with dynamic friction may also promote deformation on lower-angle structures. However, from a long-term tectonic perspective our results clearly suggest the majority of deformation during the thinning and hyperextension-exhumation phases is accommodated on high-angle normal faults that incise abandoned faults rotated to low angles.

The complex faulting patterns preserved in the final margin architecture correspond closely to structures widely observed in rifted margins[12]. From our models, we can deduce that the use of present-day observed structures to reconstruct continental extension is complicated by the non-linear character of finite-deformation (Figs. 3–5). Specifically, finite-deformation patterns at a given stage often do not intuitively connect to earlier stages due to combinations of fault segment abandonment, reactivation, rotation, intersection, and linkage to complex, non-planar structures. Indeed, a recent comparison[28] between recorded rifted margin seismic sections and seismic sections derived from a hypothesized polyphase rifting model suggest discrepancies between estimates of total extension from fault geometries largely arise from misinterpreting intersecting fault surfaces.

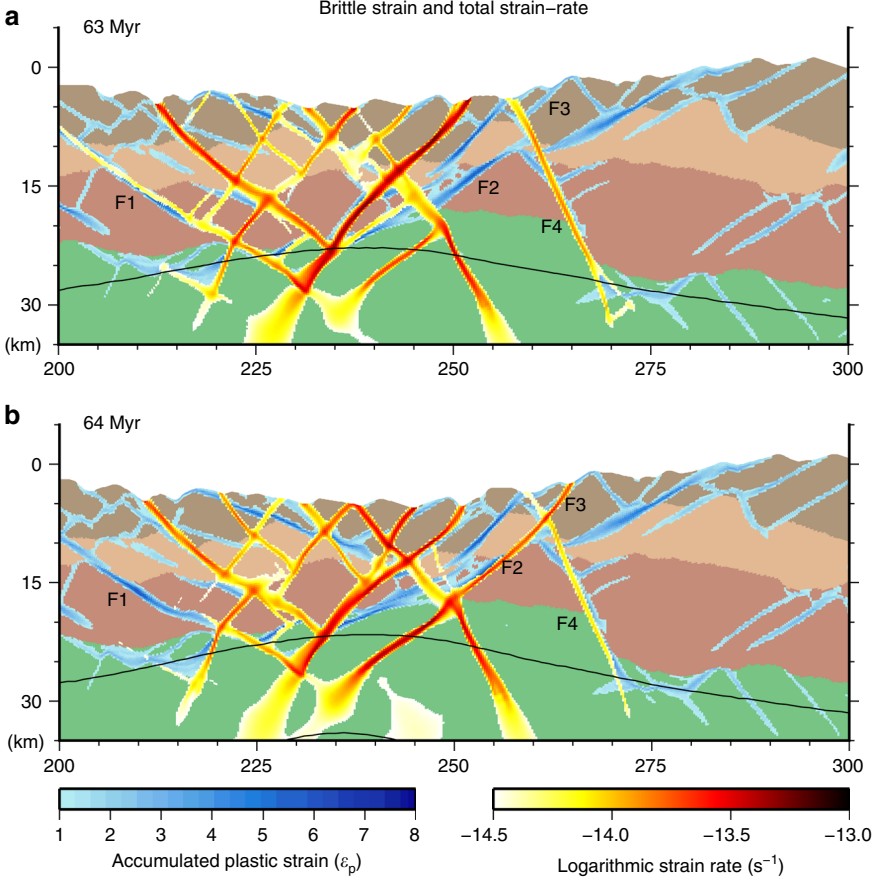

**Fig. 4** Structural interpretation of fault evolution. *Panels* show the accumulated plastic strain, total strain-rate and temperature contours plotted on top of the compositional field at **a** 63 and **b** 64 Myr. The structural interpretation follows four fault segments (F1–F4) that illustrate examples of fault deactivation, reactivation, linkage and incision (see text for more details)

Our study demonstrates that faults formed during the earliest stages of slow continental extension control the evolution of deformation coupling, mantle exhumation and breakup, and thus the formation of first- and second- order rift margin features. Rather than remaining stationary, focused zones of deformation in the thinning, hyperextension and exhumation phases (rapid extension) evolve through a complex pattern of fault linkage, abandonment, reactivation, and incision as the rift zone narrows. These findings provide new constraints on the interpretation of observed rifted margin structures and their relationship to the temporal and structural evolution of continental extension.

## Methods

**Numerical methods and governing equations**. We model extensional deformation of the lithosphere and asthenosphere with v6.0 of the finite-element, particle-in-cell code SULEC[21, 29, 30]. SULEC solves for incompressible viscous flow following the equations for conservation of mass (1) and momentum (2):

$$\nabla \cdot \mathbf{u} = 0 \tag{1}$$

$$\nabla \cdot \sigma' - \nabla P + \rho \mathbf{g} = 0 \tag{2}$$

where $\mathbf{u}$ is velocity, $\sigma'$ is the deviatoric stress tensor, $P$ is pressure, $\rho$ is density and $\mathbf{g}$ is gravitational acceleration ($g_x = 0$ and $g_y = 9.8 \, \mathrm{m \, s^{-2}}$). Density variations follow the Boussinesq approximation:

$$\rho = \rho_o (1 - \alpha(T - T_o)) \tag{3}$$

where $\rho_o$ is the reference density, $\alpha$ is thermal expansivity, $T$ is temperature and $T_o$ is the reference temperature.

Temperature ($T$) evolves according to the advection-diffusion equation:

$$\rho c \left( \frac{\partial T}{\partial t} + \mathbf{u} \cdot \nabla T \right) = \nabla \cdot k \nabla T + H + S \tag{4}$$

where $c$ is heat capacity, $k$ is thermal conductivity, $H$ is heat production and $S$ is shear heating. We use time steps of 50,000 and 10,000 years, respectively, when the applied velocity is 1 and 5 mm yr$^{-1}$. Shear heating

$$2\sigma'_{\mathrm{eff}} \dot{\varepsilon}'_{\mathrm{eff}} \tag{5}$$

is determined with the effective deviatoric stress ($\sigma'_{\mathrm{eff}}$) and strain-rate ($\dot{\varepsilon}'_{\mathrm{eff}}$), which, respectively, are defined as:

$$\sigma'_{\mathrm{eff}} = \left( \frac{1}{2} \sigma'_{ij} \sigma'_{ij} \right) \tag{6}$$

$$\dot{\varepsilon}'_{\mathrm{eff}} = \left( \frac{1}{2} \dot{\varepsilon}'_{ij} \dot{\varepsilon}'_{ij} \right) \tag{7}$$

The deviatoric strain-rate $\left( \dot{\varepsilon}'_{ij} \right)$ is calculated directly from the velocity at each time step:

$$\dot{\varepsilon}'_{ij} = \frac{1}{2} \left( \frac{\partial u_i}{\partial x_j} + \frac{\partial u_j}{\partial x_i} \right) \tag{8}$$

**Model geometry, boundary and initial conditions**. First-order features of the model geometry, boundary and initial conditions (Supplementary Fig. 1) follow the work of previous numerical studies of continental extension[13–21]. The model covers 500 and 110 km, respectively, in the horizontal and vertical directions. Outward velocities on the model sides ($V_{\mathrm{ext}}$) drive extensional deformation, while vertical inflow ($V_{\mathrm{in}}$) at the model base exactly balances material outflow such that the average surface topography remains constant through time. The model surface and base maintain fixed temperatures of, respectively, 0 and 1330 °C, while the sides are insulating boundaries (no heat flux). These thermal boundary conditions combined with the distribution of radioactive elements and variations in thermal conductivity (Supplementary Table 1) produce a geotherm characteristic of the continental lithosphere[31, 32]. The initial Moho temperature of 600 °C and surface heat flux of 55 mW m$^{-2}$ produced by this geotherm is within the range of predicted values for the continental lithosphere[32].

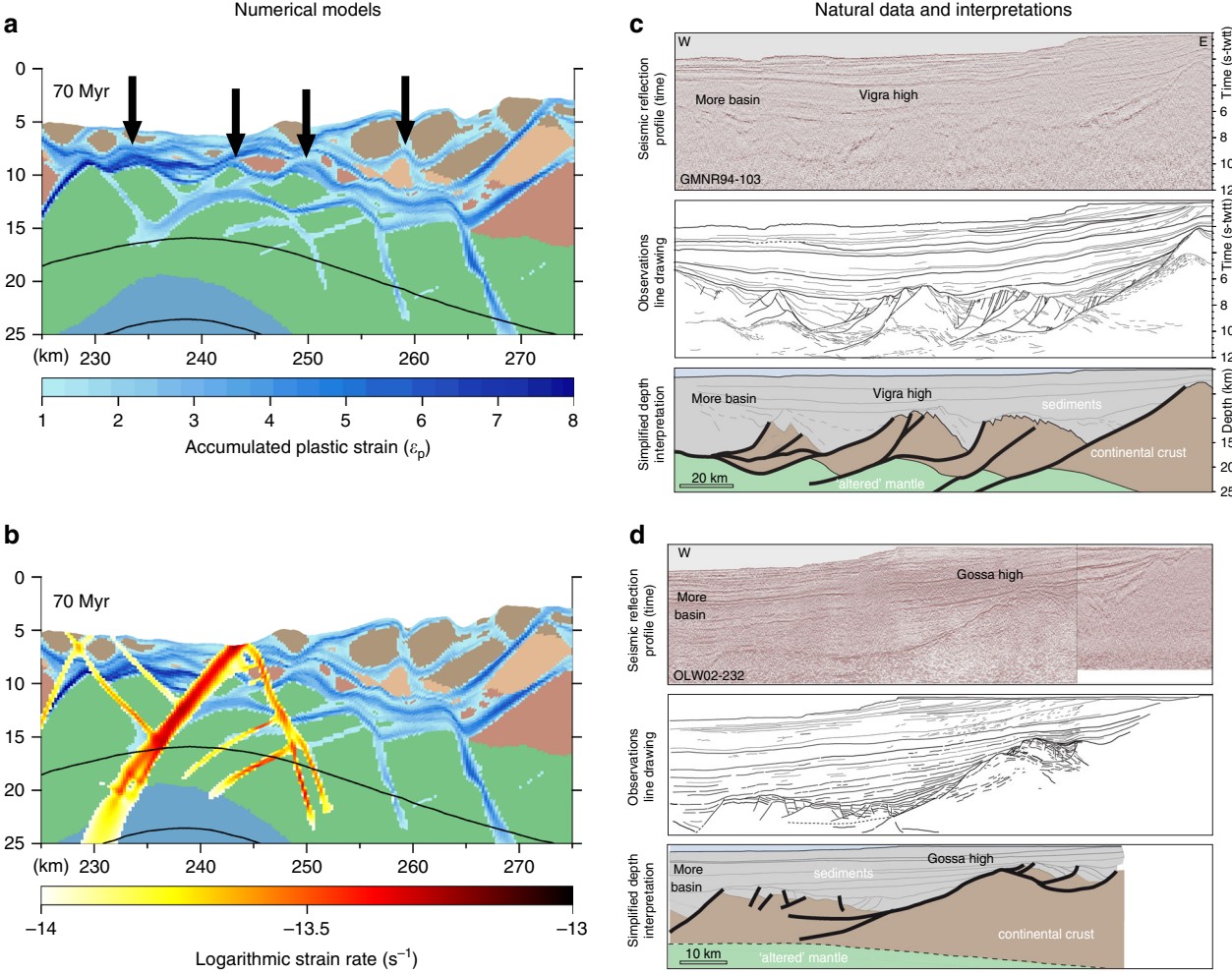

**Fig. 5** Comparison of modeled and observed key structural features. **a** Modeled lithospheric structure and accumulated deformation (plastic strain after 70 Myr). *Arrows* indicate the location of core-complex structures. **b** Strain-rate superimposed on accumulated plastic deformation illustrates high-angle incision of active faulting on low-angle (rotated) structures. **c**, **d** Structure and interpretation, modified from previous work[9], of distal margin structure along the Mid-Norwegian margin. This region illustrates examples of both core-complex geometry and incision of low-angle detachment faults by steeper faults penetrating into the mantle lithosphere. The GMNR94-103 and OLW02-232 seismic lines are released data at NPD[38]. The interpretations are modified from Osmundsen et al.[9]

**Constitutive relationships**. The lithosphere contains distinct crustal and mantle layers (Supplementary Fig. 1) with the crust subdivided into upper, middle, and lower layers. The base of the crust and lithosphere are at 40 and 100 km, respectively. Brittle and viscous constitutive relationships govern deformation, with viscous flow following a power-law model for diffusion or dislocation creep:

$$\sigma'_{\mathrm{eff}} = A^{-1/n}\dot{\varepsilon}_{\mathrm{eff}}^{1/n}d^{m/n}e^{\frac{Q+PV}{nRT}} \tag{9}$$

where $A$ is a pre-exponent, $n$ is the power-law index, $d$ is grain size, $m$ is the grain size exponent, $Q$ is activation energy, $V$ is activation volume and $R$ is the gas exponent. If both dislocation and diffusion creep are assumed to be simultaneously active, the viscosity is a composite of both creep mechanisms[33]:

$$\frac{1}{\eta_{\mathrm{comp}}} = \frac{1}{\eta_{\mathrm{diff}}} + \frac{1}{\eta_{\mathrm{disl}}} \tag{10}$$

Viscous deformation in the upper and middle crust follows a dislocation creep wet quartzite flow-law[34], while the lower crust and lithospheric mantle, respectively, follow dislocation creep models for wet anorthite[35] and dry olivine[36]. Viscous flow in the asthenosphere is a composite of dry olivine[36] diffusion and dislocation creep. Values for all flow laws are reported in Supplementary Table 1.

A Drucker–Prager yield criterion limits viscous stress by imposing an effective stress limit:

$$\sigma'_{\mathrm{eff}} = P\sin\phi + C\cos\phi \tag{11}$$

where is $\phi$ the angle of internal friction and $C$ is cohesion. $\phi$ and $C$ linearly weaken by a factor of 2[37] as total brittle strain ($\varepsilon_p$), measured as the second invariant of the brittle (plastic) finite strain tensor, accumulates between values of 0 and 0.1.

Notably, this relatively rapid rate of strain weakening helps facilitates distributed normal fault development in the initial stretching phase (Supplementary Fig. 3). Notably, structures referred to as 'faults' in the numerical simulations are in fact continuum brittle shear bands that typically span widths of 2–4 elements.

**Randomization of brittle strength**. Throughout the model, the initial internal friction angle and cohesion are, respectively, set to 20° and 20 MPa. This initial value of the element internal friction angle, however, is subsequently modified at each time step by a random perturbation factor that provides internal heterogeneities for brittle shear zones to initially localize on and in later stages simulate development of small scale weaknesses (fractures, fluid pathways) not captured by our system of equations. Such heterogeneities are pervasive in the lithosphere as a result of accumulated deformation (faulting, ductile flow, fluid–rock interaction, etc.) and exist at a wide variety of length-scales in both the crust and mantle lithosphere. The randomized internal friction angle at a given time step ($\phi^i$) is determined by the internal friction angle from the prior time-step ($\phi^{i-1}$) and a randomization factor ($F$):

$$\phi^i = \phi^{i-1} + (0.5 - r) * F \tag{12}$$

where $r$ is a random number between 0 and 1. Notably, this formulation may produce values of the internal friction angle higher than the initial value (20°), although this is unlikely in any regions that have accumulated plastic strain. More significantly, adding or subtracting the random value does not lead to excessive overall weakening of the materials and simulates that strength heterogeneities in the lithosphere this method aims to capture (fractures, inherited geological structure, fluid migration, etc.) manifest as relative changes in strength rather than only decreases. The majority of models (Figs. 2–5, Supplementary Figs. 2–4, 7–14) use a

randomization factor (*F*) of 10, while values of 2, 5, and 20 produce similar results that reflect the same underlying processes outlined in this manuscript (Supplementary Figs. 5 and 6). While the results in Figs. 2–5 and Supplementary Figs. 2–6, 9–14 use a randomization factor applied at all time steps within the simulation, restricting the number of randomization steps to various times (10, 1, or 0.1 Myr) within the stretching phase also produces the same long-term processes and very similar to nearly identical structures (Supplementary Figs. 7 and 8). These results reflect that the processes in the thinning and hyperextension-exhumation are strongly dependent on the presence of pre-existing stretching phase faults.

We have assigned a randomized friction angle to the entire thickness of the crustal and lithospheric materials in the damaged zone (Supplementary Fig. 1). However, at the initial stage of slow extension (stretching) the randomized friction perturbations within the brittle domain govern the location of the distributed faults. While inherited heterogeneities in the ductile portions of the lithosphere may certainly guide the location of faulting within the brittle domains, only assigning randomized viscous heterogeneity in the ductile domains typically leads to large amounts of crustal thinning without accompanying localized faulting. As there is no geologic evidence to support this type of deformation mode, nearly all numerical studies of lithospheric extension initialize deformation with a perturbation to the brittle field.

**Deformation focusing and initial rift width**. Strain weakening of the brittle properties varies laterally within the model. In the central 250 km of the lithosphere (Damaged Zone, Supplementary Fig. 1) the cohesion and internal friction angle weaken by a factor of $2^{37}$, while in the outer 'Strong Zones' no strain weakening occurs. These lateral variations in strain weakening focus deformation in the 'Damaged Zone'. Geologically, the outer 'Strong Zones' may be considered akin to 'stable' continental lithosphere that bounds deformed regions repeatedly reactivated during the Wilson Cycle.

**Spatial discretization**. In all models, we use elements that are linear in velocity and constant in pressure (Q1P0). The grid resolution is constant at 250 m (Figs. 2–5, Supplementary Figs. 13 and 14) or 500 m (Supplementary Figs. 2–12). Elements initially contain 16 randomly placed particles and, as deformation proceeds, injection and deletion maintains the number of particles between 9 and 36.

**Sensitivity analysis**. To validate the robustness of our results we performed a wide range of additional numerical experiments. These experiments detail the effects of numerical resolution, rates of strain weakening, brittle randomization, mantle rheology, initial stretching phase velocity, and the 'fast' phase velocity. As documented in the Supplementary Materials and main text, the key conclusions of our study are consistent across the examined parameter space.

**Data availability**. Data from this study are available from the corresponding author on request.

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

## Acknowledgements

This study was supported by the Norwegian Research Council through NFR project 213399/F20. SULEC is jointly developed by Susanne Buiter and Susan Ellis. We thank Tim Reston and Tony Lowry for insightful and highly constructive reviews.

## Author contributions

J.B.N. designed the study, ran numerical simulations, interpreted numerical data, and lead writing of the manuscript; S.J.H.B. contributed to numerical model design, data interpretation, writing of the manuscript, and is one of the developers of the software used; G.P.P. and P.T.O. provided the initial motivation for the study, guided comparisons between numerical and observed data sets, contributed to manuscript writing and co-designed figures containing natural data sets; J.T. aided in initial analysis of preliminary numerical models and writing of the manuscript.

## Additional information

**Competing interests:** The authors declare no competing financial interests.

