## [Peer Review File · Nature Communications]

Reviewers' Comments:

Reviewer #1 (Remarks to the Author)

This paper is extremely important as it presents a mechanism of crustal thinning to breakup that is quite different from other (lower resolution?) models. The key difference seems to be in how the deformation is initially localised (random weak elements rather than one specific weakness, which allows lots of early structures to form and perhaps be exploited subsequently). The paper is novel and the result extremely enlightening.

As the authors state, this leads to considerable structural complexity, which is in keeping with the observations made at rifted margins (e.g. Reston, 1995 cited, but also McDermott and Reston, *Geology*, 2015), and which many numerical modelling studies rather gloss over (more later). The comparisons between strain and strain rate (e.g. Fig 2) are particularly informative as they show which parts of existing structures are active at any one time, and as such clearly show how the active strain is focused in the middle of the older strain. In other words, the model runs also predict that extension focuses with time rather than migrates (see discussion on terminology below). The strain vs strain rate comparison also shows very clearly that the high strain rate bands only partly follow pre-existing high strain bands and in large part cross and cut the earlier structures. Thus although reactivation is important, the cross-cutting nature of the later structures is equally important.

Although I consider the results to be excellent and of interest to a wide readership, there are a number of ways that the manuscript can be tightened and improved. My main comments are suggestions of how to achieve this with minimal effort on the part of the authors.

Suggestions for improving the manuscript

Strain and strain rate

Strain rate is shown as a log scale (as is usual) which means that only reds and darkest oranges actually have significant deformation. With that in mind it is clear that deformation focuses (not migrates – see below) and that the vast majority of old structures are not significantly reactivated; the emphasis on reactivation in the text might be somewhat misleading.

It is not clear over what time interval strain rate is determined? Is it one time step in the model run and if so what is that time step? The strain rate plots tend to show conjugate structures but of course conjugate structures present well-known geometric problems, which is why one tends to become dominant. It would be interesting for instance to know if the active right dipping band in the centre top of Figure 2e, which is a new structure, not reactivating any old structures) was active mainly as an antithetic structure to the central left-dipping main band. (The right-dipping band lines up with a lower strain rate deeper old structure which may have controlled its initiation, but there is no way of knowing how long it may have been active)

How much reactivation is there?

As noted above, the comparison between the strain rate and strain shows that the majority of the older high strain zones are not active later on as extension focuses. Thus reactivation is not the principle observation. To demonstrate that there is some reactivation (parts of the active segments DO follow existing high strain bands), it would be good to be able to track a couple of faults e.g. highest strain in 2e across 2b, 2c, 2d to demonstrate reactivation as at the moment I do not see much evidence for it. Overlaying the strain rate figures on the strain figures shows that parts of the steeper high strain bands are active (not clear how much of the strain is related to this period of activity and how much quiescence there may have been – cannot have reactivation without deactivation – hence I refer to them as active old segments), so any reactivation is of

parts of different structures which is interesting and should be emphasised more clearly. The active old segments are linked by new steep structures sometimes cutting up to the surface as shortcut faults.

I suggest that in the extended data there is a figure tracking the deformation at much shorter intervals (perhaps every 1 Myr?) over critical time intervals (e.g. 60 Myr to 70 Myr) so that we can see the evolving system of faults.

Figures

The comparison with the conceptual model in Figure 1 is not very useful: it principally shows the deficiencies and oversimplifications in that particular conceptual model. A better comparison might be with models in which extension focuses leading to cross-cutting polyphase structures as that is what the model runs here demonstrate. The comparison with real seismic data is much more useful, but these comparisons need to be much clearer: how many of the modelled structures are not observed or not observable in the seismic? It would also be useful to indicate the time frame of activity of the model run's faults and fault segments in the comparison

Terminology:

Reactivation should only apply to faults which have been "deactivated" i.e. temporarily abandoned. It is not really possible to identify whether active faults (high strain rate) which follow bands of high strain (old faults) are reactivating those faults or just that those faults or fault segments have been active for a long time. The extended data figure with 1 Myr time step might help here, but some way of labelling the timing rather than just the amount of strain might be useful.

Migration implies that the locus of extension moves which is demonstrably not the case here. Although the locus of individual faulting moves to some extent (only part of the active structures are using pre-existing ones), the overall extension clearly focuses in the centre of the rift. This distinction is important as some models argue that extension migrates into unextended lithosphere. The wording throughout should be checked and in my view "migration" replaced

Tim Reston

Reviewer #2 (Remarks to the Author)

This manuscript presents an interesting study of rift deformation using a state-of-the-art numerical code and realistic rheology. The methodology appears to be sound, the results are novel, and I believe that the analysis merits publication in Nature Communications.

The main critique I would offer here is that the paper appears to "bury the lede" a bit. There are two novel results here that have significant implications in other related disciplines, made possible in part by the use of a fairly realistic approach to modeling brittle-field failure. One relates to the nature and role of low angle faulting in deformation: There is ongoing debate regarding whether low-angle detachment faults in rifted regions may be seismogenic and/or participate in the rifting process, despite the lack of seismological evidence for such events having occurred in the instrumental record. I have colleagues who are structural geologists, for example, who believe these structures are active while at low angles. However, this analysis affords a pretty convincing argument (summarized in Figures 2-3) that low-angle structures begin as high-angle structures, are rotated to low angle and then stop participating in the brittle-field shear deformation (which continues to occur on high-angle structures). This result is not even mentioned in the opening paragraph: Instead the paper opens with a relatively vague statement (lines 24-26) that "Our models demonstrate ... how faults formed in the earliest phases of continental extension control the subsequent structural evolution and complex architecture of rifted margins"... Which if anything one might interpret to mean that the low-angle faults still play a role in deformation.

The second vaguely worded result is stated as (lines 19-21) "Here, we ... show how structural

inheritance ... controls the architecture of rifted margins and their temporal evolution." More specifically, the modeling appears to show that the presence or absence of heterogeneities in brittle-field frictional strength, modeled here as randomized variations in internal frictional angle and cohesion of plastic failure, determine where and whether rift strain occurs. While it is observationally clear that reactivation of structures does occur, and it makes sense that existing faults would serve as zones of weakness in successive deformation events, one always wonders how structures get their start given Byerlee's law laboratory deformation behaviors suggesting that these properties are remarkably uniform across different rock types unless the rocks happen to be clays. The implications of this analysis would also be surprising, to me, given my intuitive expectation that heterogeneities found in the ductile domain would be more likely loci of the lithospheric weakness that seeds the extension. Stress relaxation in the ductile domain, and resulting stress focusing in the near-field brittle domain, will always precede the brittle/plastic failure. My questions would be, is the brittle-field heterogeneity a necessary feature in order to get the style of deformation observed and replicated in these models, or could similarly-distributed ductile regime heterogeneity (e.g., due to localized variations in chemistry or hydration-state) accomplish the same result? If it's the former, then the physical-state implications for the crust (i.e., that it require prior deformation events to seed it, or some process of alteration of feldspars to form clays) are important and deserve more prominent discussion than the vague sentence in the opening paragraph, and the somewhat more illuminating paragraph (lines 227-238) buried deep within the Methods section.

I will offer one other constructive criticism, and please note that I am not suggesting this needs to be addressed in any revisions of this manuscript but is simply something to bear in mind for future studies. Initial temperatures at the Moho in this model are less than 600°C (Fig. 1e). That is several hundred °C lower than is typically observed based on Pn measurements and mineral physics in the continental U.S. (e.g., Schutt et al., AGU abstract in 2016 and others), where mean Moho temperatures are ~850°C and temps are never observed below 650°C. I suspect the discrepancy has to do with your assumptions of thermal properties in the crust: heat production is relatively high at shallow depths and lower in the lower crust because radioactive elements tend to travel up-column with melts (e.g., Furlong & Chapman, *Ann. Rev. Earth Planet. Sci.*, 2013), and crustal thermal conductivity is temperature-dependent and decreases with temperature (e.g., Kukkonen et al., *Surv. Geophys.*, 1999). Both of these would tend to steepen the shallow geotherm and increase temperature at depth. Where this becomes relevant is in the ductile flow strengths for middle and lower crust, but it appears that the model offsets the low temperatures by assuming a saturated water fugacity flow law (wet quartz and wet anorthite parameterizations). This becomes a bit dodgy because the model then assumes dry olivine for the mantle (and one would expect that if the crust is hydrated, the mantle should be too). Using a more realistic higher geothermal gradient, though, the need for hydration to achieve flow in the crust is less and one could probably get roughly the same strength profile using similar (lower) water fugacities for crust and mantle.

Regards,

Tony Lowry

Response to Review #1

As the authors state, this leads to considerable structural complexity, which is in keeping with the observations made at rifted margins (e.g. Reston, 1995 cited, but also McDermott and Reston, *Geology*, 2015)

We have now cited (line 199) and referenced (#27) this publication.

The comparisons between strain and strain rate (e.g. Fig 2) are particularly informative as they show which parts of existing structures are active at any one time, and as such clearly show how the active strain is focused in the middle of the older strain. In other words, the model runs also predict that extension focuses with time rather than migrates (see discussion on terminology below).

Indeed, the term **migration** can potentially lead to multiple interpretations and we have therefore modified the text to clarify our interpretations and explanations. In essence, we effectively agree with the reviewers terminology for “rift migration” and have replaced the word “migration” in the main body of the text with different, more descriptive terminology (e.g., deactivate, reactivate, localize, evolve, etc).

The strain vs strain rate comparison also shows very clearly that the high strain rate bands only partly follow pre-existing high strain bands and in large part cross and cut the earlier structures. Thus although reactivation is important, the cross-cutting nature of the later structures is equally important.

We agree with the reviewer that reactivation and cross-cutting are both equally important and we have modified the main body of text accordingly to emphasize this point in multiple places. For example, see lines 122-124 and 145-160.

Strain rate is shown as a log scale (as is usual) which means that only reds and darkest oranges actually have significant deformation. With that in mind it is clear that deformation focuses (not migrates – see below) and that the vast majority of old structures are not significantly reactivated; the emphasis on reactivation in the text might be somewhat misleading.

We agree that our earlier text tended to over-emphasize the reactivation of faults in comparison to other processes, such as deactivation and cross-cutting, and the main body of the text has been modified to highlight the relative importance of all the key processes. In particular, we have added or modified text in reference to Figures 2-3 that highlights the timing and relative role of fault deactivation, reactivation, rotation, stretching and incision (i.e. cross-cutting).

It is not clear over what time interval strain rate is determined? Is it one time step in the model run and if so what is that time step?

In our numerical simulations strain-rate is determined from the instantaneous velocity field at each time-step. However, in the methods section we did not include an equation for the strain-rate. We have now added an equation for strain-rate to the methods section (#8, line 238).

The strain rate plots tend to show conjugate structures but of course conjugate structures present well-known geometric problems, which is why one tends to become dominant. It would be interesting for instance to know if the active right dipping band in the centre top of Figure 2e, which is a new structure, not reactivating any old structures) was active mainly as an antithetic structure to the central left-dipping main band. (The right-dipping band lines up with a lower strain rate deeper old structure which may have controlled its initiation, but there is no way of knowing how long it may have been active)

This is a good question and to provide a specific answer requires looking at an animation of the model evolution with close time intervals between (100,000 or 200,000 years). We have re-run the simulation with output every 100,000 time steps during the second fast phase of extension (the calculation time step is 10,000 years). This data is visualized within Supplementary Movie 1, which clearly shows how faults are deactivated, reactivated, incised and rotated through time. Rather than spend a significant amount of text dedicated to a particularly feature(s), we have described the general trends in more detail using Figure 2 and a few specific examples in the newly added Figure 3. If viewers wish to carefully examine the temporal evolution of the entire fault network they may view Supplementary Movie 1, which is cited multiple times in the main body of the text with Figures 2 and/or 3.

How much reactivation is there?

As noted above, the comparison between the strain rate and strain shows that the majority of the older high strain zones are not active later on as extension focuses. Thus reactivation is not the principle observation.

As noted above, we agree with this assessment and have modified our text accordingly to emphasize all of the key processes.

To demonstrate that there is some reactivation (parts of the active segments DO follow existing high strain bands) , it would be good to be able to track a couple of faults e.g. highest strain in 2e across 2b, 2c, 2d to demonstrate reactivation as at the moment I do not see much evidence for it. Overlaying the strain rate figures on the strain figures shows that parts of the steeper high strain bands are active (not clear how much of the strain is related to this period of activity and how much quiescence there may have been – cannot have reactivation without deactivation – hence I refer to them as active old segments), so any reactivation is of parts of different structures which is interesting and should be emphasised more clearly. The active old segments are linked by new steep structures sometimes cutting up to the surface as shortcut faults.

I suggest that in the extended data there is a figure tracking the deformation at much shorter intervals (perhaps every 1 Myr?) over critical time intervals (e.g. 60 Myr to 70 Myr)so that we can see the evolving system of faults.

Per the reviewer's suggestion, we have also added a new Figure (Figure 3) that superimposes the strain-rate field on top of the brittle deformation at snapshots of 63 and 64 Myr. In these two snapshots we provide a detailed structural interpretation of multiple fault strands that illustrated the processes of deactivation, reactivation and incision.

Figures

The comparison with the conceptual model in Figure 1 is not very useful: it principally shows the deficiencies and oversimplifications in that particular conceptual model. A better comparison might be with models in which extension focuses leading to cross-cutting polyphase structures as that is what the model runs here demonstrate.

On this point we disagree with the reviewer and we prefer to keep the figure unmodified. While perhaps not particularly useful for members of the continental extension community, this paper is targeting a wide-spectrum of researchers in the tectonics, structural geology, geophysics and geodynamics community. A large portion of these communities are likely not familiar with conceptual models of rifted margin formation developed in the last decade. While any conceptual model has deficiencies, the processes outlined in this particular model were the underlying motivation for the study and serves as a useful tool to compare and contrast where the numerical model diverges or agrees with specific observations.

The comparison with real seismic data is much more useful, but these comparisons need to be much clearer: how many of the modelled structures are not observed or not observable in the seismic?

We have now expanded on our discussion of the numerical-seismic data comparisons (lines 176-182) and particularly note that steep, seaward-dipping structures in the mantle are not observed in some seismic lines. Similarly, we note that in both presented seismic sections there is a lack of steep continent dipping structures, although other these features have been observed along other rifted margins (e.g. Brasil).

It would also be useful to indicate the time frame of activity of the model run's faults and fault segments in the comparison

We agree that this would indeed be incredibly useful. However, accurately quantifying such information requires extensive development of a numerical algorithm that performs the analysis in automated fashion. We are currently working on this topic, but it is not within the scope of the paper to address.

Terminology:

Reactivation should only apply to faults which have been "deactivated" i.e. temporarily abandoned.

We have adopted this terminology, which is outlined in more detail within the preceding responses.

It is not really possible to identify whether active faults (high strain rate) which follow bands of high strain (old faults) are reactivating those faults or just that those faults or fault segments have

been active for a long time. The extended data figure with 1 Myr time step might help here, but some way of labelling the timing rather than just the amount of strain might be useful.

The new Figure 3 and Supplementary Movie 1 do indeed aid in identifying new faults versus reactivated faults.

Migration implies that the locus of extension moves which is demonstrably not the case here. Although the locus of individual faulting moves to some extent (only part of the active structures are using pre-existing ones), the overall extension clearly focuses in the centre of the rift. This distinction is important as some models argue that extension migrates into unextended lithosphere. The wording throughout should be checked and in my view “migration” replaced

We agree with this definition of the terminology and replaced the term migration with other, more specific terms throughout the text.

Response to Review #2

The main critique I would offer here is that the paper appears to “bury the lede” a bit. There are two novel results here that have significant implications in other related disciplines, made possible in part by the use of a fairly realistic approach to modeling brittle-field failure.

We address each of these points below.

One relates to the nature and role of low angle faulting in deformation: There is ongoing debate regarding whether low-angle detachment faults in rifted regions may be seismogenic and/or participate in the rifting process, despite the lack of seismological evidence for such events having occurred in the instrumental record. I have colleagues who are structural geologists, for example, who believe these structures are active while at low angles. However, this analysis affords a pretty convincing argument (summarized in Figures 2-3) that low-angle structures begin as high-angle structures, are rotated to low angle and then stop participating in the brittle-field shear deformation (which continues to occur on high-angle structures). This result is not even mentioned in the opening paragraph: Instead the paper opens with a relatively vague statement (lines 24-26) that “Our models demonstrate ... how faults formed in the earliest phases of continental extension control the subsequent structural evolution and complex architecture of rifted margins” ... Which if anything one might interpret to mean that the low-angle faults still play a role in deformation.

In the case of low-angle normal faults, we did indeed “bury the lede” somewhat as our results are not very applicable to the debate on whether low angle normal faults are seismically active. Our results shed light on crustal- and lithospheric processes on time-scales 10^5 - 10^7 years that arise from ductile flow and brittle shear zones with widths ranging from 500-1000 meters. In contrast, shedding light on the seismicity of low angle normal faults requires taking into account stick-slip constitutive relationships (velocity weakening/strengthening), fluid migration and associated changes in pore fluid pressure, elastic loading and material acceleration.

That said, we have noted that our results support one hypothesis (added citation # 26) for the formation of low angle faults. But we also emphasize that our results should not be over interpreted within the context of the debate over low angle normal fault seismicity (lines 184-191).

The second vaguely worded result is stated as (lines 19-21) “Here, we ... show how structural inheritance ... controls the architecture of rifted margins and their temporal evolution.” More specifically, the modeling appears to show that the presence or absence of heterogeneities in brittle-field frictional strength, modeled here as randomized variations in internal frictional angle and cohesion of plastic failure, determine where and whether rift strain occurs. While it is observationally clear that reactivation of structures does occur, and it makes sense that existing faults would serve as zones of weakness in successive deformation events, one always wonders how structures get their start given Byerlee’s law laboratory deformation behaviors suggesting that these properties are remarkably uniform across different rock types unless the rocks happen to be clays. The implications of this analysis would also be surprising, to me, given my intuitive expectation that

heterogeneities found in the ductile domain would be more likely loci of the lithospheric weakness that seeds the extension. Stress relaxation in the ductile domain, and resulting stress focusing in the near-field brittle domain, will always precede the brittle/plastic failure. My questions would be, is the brittle-field heterogeneity a necessary feature in order to get the style of deformation observed and replicated in these models, or could similarly-distributed ductile regime heterogeneity (e.g., due to localized variations in chemistry or hydration-state) accomplish the same result? If it’s the former, then the physical-state implications for the crust (i.e., that it require prior deformation events to seed it, or some process of alteration of feldspars to form clays) are important and deserve more prominent discussion than the vague sentence in the opening paragraph, and the somewhat more illuminating paragraph (lines 227-238) buried deep within the Methods section.

The reviewer brings up an excellent point above, although we feel it is not within the scope of this publication to fully address. The short answer is that in order for brittle deformation to localize relatively quickly some form of weak heterogeneity in the brittle domain is required. Without a weak heterogeneity in the brittle domain the lithosphere tends to undergo bulk thinning and localized deformation (faults) may not develop until after significant thinning (25-50%) of the crust. To our knowledge, there are no observations of crustal thinning without some form of related brittle deformation. Our reasoning for using randomized weak heterogeneities in the brittle field is now discussed in the methods section (lines 309-318).

A slightly longer answer is that earlier tests we did show that these randomized brittle heterogeneities are essential for distributed extension as a single heterogeneity (‘seed’) immediately localizes extensional deformation along two main conjugate shear zones (see Supplementary Figure 2). A seed in the ductile domain also has this effect. However, any localization, distributed or not, requires a certain strength contrast between heterogeneities and their surrounding and such contrasts may be difficult to achieve in the purely ductile domain, implying that the model might not ‘see’ ductile heterogeneities enough to localize. We do not discuss how variations in the location or type (purely mechanical, mechanical + thermal) of the randomized (or single) seed(s) affect continental extension patterns as this topic is in fact the

focus of a study by one of the authors, forming the basis of an extensive publication on the subject. However, we have added two additional supplementary Figures (7-8) that further elucidate our method for generating distributed normal faulting through random heterogeneities.

While the reviewer certainly has brought up an interesting point, we feel there is not sufficient room in the publication to properly address this topic as it would require incorporating and running dozens of additional models. For the purposes of this manuscript, we feel that our methodology for generating distributed normal faults alone will be a significant contribution to the modeling community.

I will offer one other constructive criticism, and please note that I am not suggesting this needs to be addressed in any revisions of this manuscript but is simply something to bear in mind for future studies. Initial temperatures at the Moho in this model are less than 600°C (Fig. 1e). That is several hundred °C lower than is typically observed based on Pn measurements and mineral physics in the continental U.S. (e.g., Schutt et al., AGU abstract in 2016 and others), where mean Moho temperatures are ~850°C and temps are never observed below 650°C. I suspect the discrepancy has to do with your assumptions of thermal properties in the crust: heat production is relatively high at shallow depths and lower in the lower crust because radioactive elements tend to travel up-column with melts (e.g., Furlong & Chapman, *Ann. Rev. Earth Planet. Sci.*, 2013), and crustal thermal conductivity is temperature-dependent and decreases with temperature (e.g., Kukkonen et al., *Surv. Geophys.*, 1999). Both of these would tend to steepen the shallow geotherm and increase temperature at depth. Where this becomes relevant is in the ductile flow strengths for middle and lower crust, but it appears that the model offsets the low temperatures by assuming a saturated water fugacity flow law (wet quartz and wet anorthite parameterizations). This becomes a bit dodgy because the model then assumes dry olivine for the mantle (and one would expect that if the crust is hydrated, the mantle should be too). Using a more realistic higher geothermal gradient, though, the need for hydration to achieve flow in the crust is less and one could probably get roughly the same strength profile using similar (lower) water fugacities for crust and mantle.

To ensure the lithospheric mantle rheology does not have a first-order effect on our results, we ran a simulation with a wet olivine mantle flow law assigned to the lithospheric mantle. The results of this simulation are displayed in Supplementary Figures 11-12, which can be compared to Supplementary Figures 9-10 and Figures 1-4. The results indicate that the main effect of a weaker mantle is to delay the onset/end of each phase of deformation slightly, but the key processes and observations highlighted in this study are still present.

We agree with the reviewer that our choice of the distribution of radiogenic heating elements and Moho temperatures can be improved for future studies, although we note a Moho temperature of 600 °C is a commonly assumed value in numerical modeling studies for a 40 km thick crust (see new reference #37).

Following the reviewer's comments, we re-ran simulations with a Moho temperature of 700 °C or a variable amount of radiogenic heating in the crust. As we expected, this increase in the geothermal gradient places the mantle within the ductile regime for the assigned stretching phase

velocity (1 mm yr^{-1}). As a result, brittle faults do not form in the mantle lithosphere and the transition to coupling between crust and mantle deformation can be delayed by 10's of Myr even after the onset of faster velocities (5 or 10 mm yr^{-1}). This delay significantly alters the modeled structures and can lead to "a second" stretching phase with extensive stretching of the upper and middle crust. As this process is not observed along passive margins, we believe an additional form of mechanical randomization or weak seed needs to be added to the mantle lithosphere in cases where the uppermost mantle is not in the brittle regime (and faults do not form). Alternatively, passive margins may simply have a lower initial geothermal gradient at the onset of extension. The fact that our results reproduce many of the key features of passive margin structure suggests that portions of the mantle lithosphere are likely to be undergoing brittle deformation at the onset of extension.

A second way to induce brittle deformation in the mantle lithosphere with a higher geothermal gradient is to use a higher initial stretching phase velocity. While this type of parameter exploration is certainly warranted for future studies, we note that many regions of continental extension have initially low extension rates, which only ramp up towards continental breakup (see cited paper: Brune et al., Nature, 2016). As the reviewer did not request we address his point directly in this paper, we will leave this work to a future study. As noted above, one of the co-authors is in fact preparing a separate manuscript that directly addresses the issue of how to initiate deformation under different initial conditions.

Reviewers' Comments:

Reviewer #1:

Remarks to the Author:

Review of resubmission of Naliboff et al

This is a strong and very interesting paper, supported by some excellent figures. Whereas some modelling seems oversimplistic and lacking the details that are likely to prove crucial in the real world, the modelling presented here highlights how important those details are, and how structures interact. The new Figure 3 is excellent: plotting strain and strain rate together shows very clearly how some structures are reactivated and other cut across, giving, in my opinion, some very interesting insights into the sort of processes that really do seem to occur at rifted margins. This figure shows some of the process that is occurring in the models, and probably occurs in the real world, far better than I have seen before in numerical modelling.

My, generally minor, concerns have all been dealt with, and I consider that this paper is ready for publication. I recommend it is accepted as it is.

Reviewer #2:

Remarks to the Author:

I remain supportive of publication of this manuscript in Nature Communications. My one remaining concern has to do with Lines 184-191: I can understand (and respect) the authors' reticence to wade into a debate that their modeling wasn't really designed to address, but it seems to me that this paragraph as stated under-interprets the results. "Seismogenic" as discussed in the referenced Axen paper can have two (albeit not very clearly demarked) end-member meanings: (1) capable of rupturing, or (2) capable of generating large-magnitude, strain-dominating events that should be considered as part of seismic hazard (exemplified in Axen-07 with a citation of speculation in Wernicke-95). It is fairly clear from recent seismic and geodetic studies that the first is possible and may even be likely; large magnitude events from Denali to Kaikoura have activated multiple subevents on faults at high angles (and in some cases unconnected) to the structure that released the majority of the strain moment. That interpretation is certainly consistent with all of the examples of observations supporting LANF slip cited in the Axen paper. But the modeling performed here would suggest that slip on low angle structures cannot be a dominant mode of moment release. If the roughly ten orders of magnitude of weakening of the static frictional yielding parameter that this model permits is not enough to favor deformation on the low-angle structures over the formation of new deformation zones, then pore fluid pressures or rate-state frictional dynamics on a fault at scales below the modeling threshold are unlikely to change that.

I understand the authors' point that neglecting elements of the frictional rheology and fault fluids leaves some wiggle-room for interpreting whether or not low angle normal faults may be seismogenic, and it's certainly a fair concern that is raised regarding the differences in temporal and spatial scales of the model from those that are necessary for elastodynamic modeling—But arguing that unmodeled processes make these modeling results irrelevant to the seismogenesis debate, without drawing the distinction between subevent seismogenesis and strain-dominant seismogenesis, risks weakening the other interpretations of the paper. This paragraph could be interpreted as an argument that unmodeled processes may completely change the strain behavior to favor moment-dominant slip, hence high longer-term strain rates, on structures that in this model are abandoned, thus potentially invalidating the model. Since the model performs rather well in replicating other structural observations, it seems to me the modeling also argues against the moment-dominant end-member of LANF slip. But, I leave it to the authors as to how or whether to address this in the paper.

Other changes to the manuscript are helpful, including those in response to the other reviewer,

and my earlier concerns have been addressed adequately. The clarification of the need for brittle-domain heterogeneity to quickly localize brittle-field structures is interesting. It's worth noting that concerns that ductile-domain strength heterogeneities may not be large enough to generate contrasts needed to localize deformation may be ill-founded, given that hydration state can readily vary ductile strength by many orders of magnitude (but it sounds as though the scale on which the strength variation occurs may be important also for localization? In which case, we may be talking about different things when we say localization, because my review referred to lithospheric-scale localization introduced by allowing for brittle-field strain weakening only within the central 250 km of the model-- That is the localization that I would find more plausible if it were seated in the ductile rather than the brittle-field. As the response to my review appears to be discussing localization of shear zones, it's worth noting that orders-of-magnitude variations in ductile strength associated with hydration state are unlikely on scales of less than a few km, given the diffusivity coefficients in the weaker materials).

Regards,

Tony

Response to Review #1

This is a strong and very interesting paper, supported by some excellent figures. Whereas some modelling seems oversimplistic and lacking the details that are likely to prove crucial in the real world, the modelling presented here highlights how important those details are, and how structures interact. The new Figure 3 is excellent: plotting strain and strain rate together shows very clearly how some structures are reactivated and other cut across, giving, in my opinion, some very interesting insights into the sort of processes that really do seem to occur at rifted margins. This figure shows some of the process that is occurring in the models, and probably occurs in the real world, far better than I have seen before in numerical modelling.

My, generally minor, concerns have all been dealt with, and I consider that this paper is ready for publication. I recommend it is accepted as it is.

As reviewer #1 has not raised any additional comments or critiques to address, our manuscript has only been modified to address the comments of reviewer #2. However, we would like to thank reviewer #1 again for his insightful and constructive review that improved the quality, clarity and accuracy of the manuscript.

Response to Review #2

I remain supportive of publication of this manuscript in Nature Communications. My one remaining concern has to do with Lines 184-191: I can understand (and respect) the authors' reticence to wade into a debate that their modeling wasn't really designed to address, but it seems to me that this paragraph as stated under-interprets the results. "Seismogenic" as discussed in the referenced Axen paper can have two (albeit not very clearly demarked) end-member meanings: (1) capable of rupturing, or (2) capable of generating large-magnitude, strain-dominating events that should be considered as part of seismic hazard (exemplified in Axen-07 with a citation of speculation in Wernicke-95). It is fairly clear from recent seismic and geodetic

studies that the first is possible and may even be likely; large magnitude events from Denali to Kaikoura have activated multiple subevents on faults at high angles (and in some cases unconnected) to the structure that released the majority of the strain moment. That interpretation is certainly consistent with all of the examples of observations supporting LANF slip cited in the Axen paper. But the modeling performed here would suggest that slip on low angle structures cannot be a dominant mode of moment release. If the roughly ten orders of magnitude of weakening of the static frictional yielding parameter that this model permits is not enough to favor deformation on the low-angle structures over the formation of new deformation zones, then pore fluid pressures or rate-state frictional dynamics on a fault at scales below the modeling threshold are unlikely to change that.

In the reviewer statement above, we have underlined the key portion of the statement that we wish to address. We agree that our results clearly show that in the “fast” stages of continental rifting (thinning, hyperextension-exhumation) the majority of deformation is accommodated on high-angle structures incising faults that have been rotated to low-angles. If our models correctly capture the physics of long-term tectonic processes, then by definition our results should imply that the dominant mode of moment release (seismic time scales) should be on high-angle structures as opposed to low-angle structure. Given that we are still missing some of the physics and numerical methods required to model the short-term earthquake cycle as part of long-term tectonics, we have not made this statement in the text. To make this statement we still feel additional modeling efforts are required. We hope that our modifications to the text, recopied below in italics, have produced an appropriate balance between emphasizing the importance of our results with regards to the low-angle normal fault debate, while not overstating their application to seismic processes.

Discussion paragraph modified to revise our conclusions with regards to the relationship between the model results and low-angle normal fault seismicity:

The process of fault rotation and abandonment outlined in our experiments at first-order matches one proposed model for low-angle normal fault development²⁵. However, interpretation of these results with respect to the debate on low-angle normal fault seismicity²⁶ should be considered within the context of our numerical assumptions. For example, weakening the friction angle to lower values^{14,16}, including elastic effects²⁷ or the presence of fluids²⁶ may also promote accommodation of deformation on lower angle structures. Similarly, higher numerical resolutions and constitutive relationships associated with dynamic friction may also promote deformation on lower-angle structures. However, from a long-term tectonic perspective our results clearly suggest the majority of deformation during the thinning and hyperextension-exhumation phases is accommodated on high-angle normal faults that incise abandoned faults rotated to low angles.

I understand the authors' point that neglecting elements of the frictional rheology and fault fluids leaves some wiggle-room for interpreting whether or not low angle normal faults may be seismogenic, and it's certainly a fair concern that is raised regarding the differences in temporal and spatial scales of the model from those that are necessary for elastodynamic modeling—But arguing that unmodeled processes make these modeling results irrelevant to the seismogenesis

debate, without drawing the distinction between subevent seismogenesis and strain-dominant seismogenesis, risks weakening the other interpretations of the paper. This paragraph could be interpreted as an argument that unmodeled processes may completely change the strain behavior to favor moment-dominant slip, hence high longer-term strain rates, on structures that in this model are abandoned, thus potentially invalidating the model. Since the model performs rather well in replicating other structural observations, it seems to me the modeling also argues against the moment-dominant end-member of LANF slip. But, I leave it to the authors as to how or whether to address this in the paper.

Again, we agree with the reviewer on this point. If neglecting un-modeled processes such as fluid migration, dynamic friction constitutive relationships and other factors make our models wholly inapplicable to short-term styles of deformation, then they are unlikely to accurately capture longer-term styles of deformation that are simply the integrated sum of the shorter-term processes. As noted in above, we have modified the discussion to state that our results clearly support the majority of deformation being accommodated on high-angle structures incising faults rotated to lower-angles.

Other changes to the manuscript are helpful, including those in response to the other reviewer, and my earlier concerns have been addressed adequately. The clarification of the need for brittle-domain heterogeneity to quickly localize brittle-field structures is interesting. It's worth noting that concerns that ductile-domain strength heterogeneities may not be large enough to generate contrasts needed to localize deformation may be ill-founded, given that hydration state can readily vary ductile strength by many orders of magnitude (but it sounds as though the scale on which the strength variation occurs may be important also for localization? In which case, we may be talking about different things when we say localization, because my review referred to lithospheric-scale localization introduced by allowing for brittle-field strain weakening only within the central 250 km of the model-- That is the localization that I would find more plausible if it were seated in the ductile rather than the brittle-field. As the response to my review appears to be discussing localization of shear zones, it's worth noting that orders-of-magnitude variations in ductile strength associated with hydration state are unlikely on scales of less than a few km, given the diffusivity coefficients in the weaker materials).

We apologize for misinterpreting this point during the first round of review and we completely agree that on large scales (100's of km) strength heterogeneities in the ductile domain (lower crust, mantle lithosphere) almost certainly play a large role in localizing deformation. As mentioned previously, multiple authors on this manuscript are in fact actively exploring the role of initiating extension with thermal or mechanical heterogeneities in the brittle versus viscous domain. In our view, broad deformation likely initiates along or in major lithospheric boundaries that contain significant strength variations within the crust and mantle lithosphere. Indeed, observations suggest that many continental rifts initiate on old suture zones. Given their complex deformation history, the ductile domains inevitably contain large-scale strength contrasts. While variations in brittle strength alone were enough to localize deformation in the models presented here, future work is exploring combining strength heterogeneity in the brittle and viscous domains.

Again, we would like to thank the reviewer for the constructive and insightful reviews that helped to significantly improved our manuscript.

In particular, we would like to thank the reviewer for raising the issue of linking short-term deformation patterns to long-term tectonic deformation. This link is often neglected in long-term tectonic discussions and moving forward it should be at the fore-front of discussion within the geodynamics community.